

# Relationship between age, category and experience with the soccer referee's self-efficacy

José López Aguilar[1], Alfonso Castillo-Rodriguez[1], José L. Chinchilla-Minguet[2] and Wanesa Onetti-Onetti[3]

[1] Department of Physical Education and Sports, University of Granada, Granada, Spain
[2] Department of Didactics of Languages, Arts and Sport, University of Malaga, Malaga, Spain
[3] Faculty of Education, Universidad Internacional de la Rioja. UNIR, Logroño, Spain

## ABSTRACT

Soccer referees (SRs) encounter stressful situations during competitions and sometimes even outside them, which may affect their decision making. Therefore, it is important that they possess or acquire optimal levels of self-efficacy, since it is related to less stress during competition, also guaranteeing sports performance and prevent sports abandonment. The objectives of this study were to characterize the profile, in terms of self-efficacy, of SRs depending on their category, age, and experience and to determine the relationship of these factors on SR self-efficacy. Two-hundred fifty-six Spanish referees participated in this study and Referee Self-Efficacy Scale was administered and completed. The results indicated that the SRs older than 25 years, of national category, and with experience greater than or equal to 8 years, have higher levels of self-efficacy than those with the least ($p < .01$). Likewise, moderate positive correlations were also observed between global self-efficacy and the category, age, and experience of the SRs. In conclusion, age, category and experience factors relate the self-efficacy of the SR, which can explain up to 17% of the variance, affecting decision-making and other decisive behaviors in the competition. These findings are of interest to delegations and referee committees seeking to implement psychological intervention programs to prevent burnout and abandonment of sports practice due to the consequences of low self-efficacy.

## INTRODUCTION

Sports referees and judges are essential in competition since they ensure that sports games is carried out safely and fairly with respect to the regulations of the sport in question (*Warner, Tingle & Kellett, 2013*). Although the referee has an indispensable role in the correct development of the indicated sporting discipline, refereeing has scarcely been studied by researchers (*Guillén et al., 2019*; *Loghmani, Taylor & Ramzaninejad, 2018*; *Webb et al., 2016*). Referees must be attentive to all aspects of the game, evaluate situations, and make decisions in a timely manner (*Schweizer et al., 2011*). There are more than 5 million soccer referees (SRs) around the world (*FIFA, 2004*) making decisions and enforcing the

Corresponding authors
Alfonso Castillo-Rodriguez, acastillo@ugr.es
José L. Chinchilla-Minguet, jlchinchilla@uma.es

rules of the game (*Soriano Gillué et al., 2018*) as well as paying attention to the behavior of soccer players (both on the field and on the bench) and technical staff (*Castillo-Rodríguez, Muñoz-Arjona & Onetti-Onetti, in press*). Proper decision making depends on multiple factors, such as, optimal physical condition for its category and having passed physical tests and theoretical tests established by the corresponding Referees Committee (*Muñoz-Arjona & Castillo-Rodríguez, 2020*). The SR must be able to run distances similar to that of soccer players in their categories (*Barros et al., 2007*; *Di Salvo et al., 2007*) to have better positioning in the plays and therefore a clearer observation to make timely decisions. In addition, psychological skills such as concentration, self-confidence, self-efficacy, motivation (*Castillo-Rodríguez, Muñoz-Arjona & Onetti-Onetti, in press*), among others related to sports performance, are required that allow them to make judgments correctly or face erroneous decision-making due to the continuous stress of competition (*Soriano Gillué et al., 2018*) to achieve excellent refereeing (*Giske, Hausen & Johansen, 2016*; *Weinberg & Richardson, 1990*). Today, the stress of competition is greater due to the social and economic impact that soccer has (*Ramírez et al., 2006*), since a decision can cause variations in the classification and consequently, the budget of said team to the following season. If this stress affects the SR during the competition, it could impact on the physiological level (*González-Oya, 2006*), e.g., increase in the minimum heart rate at the beginning of the competition which may impair SR performance (*Castillo-Rodríguez, Muñoz-Arjona & Onetti-Onetti, in press*) and alterations in motor and visual perception (*Tornero-Aguilera & Clemente-Suárez, 2018*). These physiological changes could negatively affect judgment of a certain action in the competition. For these reasons, the development of psychological skills becomes important if they are practiced for a considerable time together with physical training (*González-Oya & Dosil, 2004*; *González-Oya & Dosil, 2007*; *Guillén & Feltz, 2011*; *Ramírez et al., 2006*).

A psychological construct that has been shown to reduce stress and anxiety related to the performance of a task or job is that of self-efficacy (*Bandura, 1997*). The concept of self-efficacy of a person includes optimistic beliefs in himself and it is defined as the strength of conviction of a person to successfully perform a behavior required to achieve a certain result in a task (*Bandura, 1997*). These perceptions are expected to influence the choice of tasks, effort made and resistance to failure or better, resilience. Furthermore, it is known that self-efficacy allows people to adapt effectively to new and changing situations (*Callan, Terry & Schweitzer, 1994*) as would occur in the case of the SR or sports judge. According to the theory of self-efficacy, it influences stress and anxiety through beliefs about personal control of actions, thoughts and affection (*Bandura, 1997*). Consequently, if the SRs have greater self-efficacy, they will have greater self-confidence and will be able to handle the matches better since they will not have high levels of stress impairing their performance during matches. It should be noted that researchers have developed, based on *Bandura (1997)*, different conceptual frameworks to differentiate self-efficacy in different areas such as organizational (*Stajkovic & Luthans, 1998*), academic (*Federici & Skaalvik, 2012*; *Bong, 2001*) and sports (*Sullivan & Kent, 2003*). Within the sports self-efficacy, there is the one related to the referee, which is called refraction for convenience. Highly effective referees must be more precise in their decisions, more effective in their performance,

more committed to their profession, have more respect from coaches, administrators and other officials, and be able to avoid the stress that refereeing generates. In fact, the aspect that most interests and worries referees is self-confidence, as some empirical studies have confirmed (*Guillén & Jimenez, 2001*; *Guillén, 2003a*).

In the last years, studies have been carried out on the construct of self-efficacy in different areas such as academic or sports, in which we highlight studies with athletes and coaches. This is not the case at the referee level because being one of the most worrying aspects, this field of knowledge is scarcely studied by science (*Ede, Hwang & Feltz, 2011*). Within referee topic, self-efficacy is the confidence or belief of successful decision-making by the referee (*Guillén & Feltz, 2011*). One way to validly and reliably measure self-efficacy in referees and sports judges is through the REFS questionnaire (Referee Self-Efficacy Scale) developed and validated in its original version by *Myers et al. (2012)* and in its Spanish version by *Guillén et al. (2019)*. In addition to providing a one-dimensional score, it also includes multidimensionality with four specifics in refereeing: game knowledge, which refers to the confidence that the SR has in the knowledge of his or her sport/regulations; decision making refers to the SR's confidence and ability to make decisions during the game; pressure refers to the SR's confidence in not being influenced by the pressure of the encounter; and communication which is the referee's ability to communicate effectively. We must emphasize and consider that the subscales of game knowledge, decision making, communication, and preassure, which are considered psychological abilities, are measured in the REFS questionnaire.

Regarding studies on self-efficacy in SR, little treatment has been observed in this field (*Guillén et al., 2019*), hence there is a need to investigate self-efficacy in referees to determine its influence on sports performance. A study carried out on handball referees stands out (*Diotaiuti et al., 2017*); it showed that referees of a higher category (national referees) had higher levels of self-efficacy than referees of a lower category (regional referees). This category is also recognized in science for the level of the referees, since they must pass some physical tests in order to compete in higher categories (*Muñoz-Arjona & Castillo-Rodríguez, 2020*). Similarly, experience in refereeing is also influential in self-efficacy, since referees with less than 4 years of experience had lower levels of self-efficacy than referees with more experience. However, no differences were found for age in terms of confidence between other psychological abilities in the study by *Nazarudin et al. (2014)* in rugby referees; this may be due to the high age of the sample, as only 25% of the subjects were under 30 years of age. As mentioned before, the higher-category SRs generally exhibit higher levels of self-efficacy (*Diotaiuti et al., 2017*), enabling mitigation and reduction of stress during matches (*Guillén, 2003b*) and increased use of mental training strategies at similar levels of physical training (*Giske, Hausen & Johansen, 2016*). However, other studies reported a negative correlation between self-efficacy and stress (*González-Oya & Dosil, 2004*; *Soriano Gillué et al., 2018*), with the least experienced SRs facing the greatest stress in competition. In addition, another study (*Micoogullari et al., 2017*) showed that SRs from Turkey with more than 15 years of experience had greater stress control than those with less experience. Regarding age and category, *González-Oya & Dosil (2004)* demonstrated that older SRs (over 40 years old) have greater stress control than younger ones (less than 20 years old). In

addition, the SRs of the highest category obtained higher stress control scores, which could mean that the SRs of older age and category have greater self-efficacy than those of the lower category. In soccer, basketball and handball sports, the Turkish referee's self-efficacy is positively related to age and experience (*Karacam & Adiguzel, 2019*; *Karaçam & Pulur, 2017*), presenting variability depending on the type of sport, being in handball referees, the lowest scores (*Karaçam & Pulur, 2017*). However, it is unknown to date if these results would be the same in referees from other countries, and therefore, who compete in other leagues with different cultures, such as Spain. The contribution of the present study consists in demonstrating that the self-efficacy of the Spanish SR increases according to the category (Hypothesis [H1a]), age (H1b) and experience (H1c), and therefore could be determining factors for the SR to be able to rise in category, through linear regressions (H2). Likewise, the aims of this study were to characterize the self-efficacy of SRs depending on their category, age, and experience, and to establish relationships of these factors in order to finally establish predictive equations of perceived self-efficacy. In this way, the importance of the study resides in better understanding the psychological profile of SR in order to prevent anxiety and stress states derived from low self-efficacy.

## METHOD

### Participants

Two-hundred fifty-six male amateur SRs belonging to the Andalusia Committee of Soccer Referees voluntarily participated in the present study. The sampling was of the non-probabilistic type from the Andalusia SR population. After contacting the Andalusia Committee, they communicated it to the SR. The inclusion criteria were that the referees had passed the corresponding relevant physical and theoretical tests of the season and that they had not had injuries in the last 6 months that would have prevented them from carrying out their refereeing work. Of these two-hundred fifty-six amateur SRs, 72 belong to national category (highest category), 50 SRs of Honor Division or State category, 78 SRs of Provincial category, and 56 are SRs of Base category (lowest category). The ages of the SRs were between 18 and 34 years old; the weight between 51 and 105 kg; height between 160 and 191 cm; and experience from 1 to 16 seasons in refereeing. The mean age, weight, height, and experience in refereeing were $23.7 \pm 3.4$ years, $72.7 \pm 8.9$ kg, $177.5 \pm 8.9$ cm, and $6.45 \pm 3.5$ years of refereeing experience, respectively. Self-efficacy data were collected in March 2020. Participants and Committee were informed of the study procedures, objectives, methodology, benefits, and potential risks. This study was approved by the Ethics Committee of the University of Granada (471/CEIH/2018).

### Instruments

First, an socio-demographic ad-hoc test was established to collect data from each referee corresponding to age, height, weight, refereeing experience, category, and injuries in the last 6 months. Second, the REFS (Referee Self-Efficacy Scale) questionnaire was administered in its Spanish version (*Guillén et al., 2019*); it was initially developed by *Myers et al. (2012)* and contains 13 items. It is a Likert scale of scoring from 1 to 5, with 1 being very low confidence and 5 being the highest confidence in their abilities. The questionnaire

**Table 1** Normality and reliability (Cronbach's α) of the self-efficacy dimensions (REFS questionnaire).

|  | Normality | Reliability |
| --- | --- | --- |
| GK | .053[*] | .776 |
| DM | .040[*] | .794 |
| PR | .088[*] | .745 |
| CO | .034[*] | .766 |
| Σ Self-efficacy | .019[*] | .883 |

Notes.

[*]$p \geq .200$.

GK, Game knowledge; DM, Decision making; PR, Pressure; CO, Communication; Σ Self-efficacy, Global self-efficacy.

differentiates 4 dimensions: game knowledge (e.g., "Understand the basic strategy of the game", items 1–3), decision making (e.g., "Make critical decisions during competition", items 4–6), pressure (e.g., "Uninfluenced by pressure from players", items 7–9) and communication (e.g., "Communicate effectively with coaches", items 10–13). Regarding reliability, Cronbach's alpha coefficient in terms of global self-efficacy was .883; in terms of constructs, reliability indexes between .745 and .794 were obtained, with pressure and communication being the lowest and highest indexes, respectively (Table 1). To ensure the validity of the instrument, the questions and concepts to Promote Transparent Reporting of Measurement Practices (*Flake & Fried, 2020*) were taken into consideration. The constructs have been defined and the theories that support them described in the introduction. The selection of the measure has been justified, as well as the consideration of respecting the criteria for the administration procedure, the scores and the treatment, without modifying any of the test questions.

## Design and Procedure

This research was a descriptive cross-sectional study of a single sample collection. It took place in March 2020, before the suspension of the competition by COVID-19 happened. Prior to this, in September 2019, conversations were held with the Andalusian Committee of Football Referees to accept that the study be carried out. A report was sent with the objectives, instruments and actions carried out with the referees Later (in January 2020), and with the help of these organizations, we contacted the SRs to explain the objectives, methodology, and research protocols. In addition, each subject provided us written informed consent. Subsequently, the questionnaire was sent to the SRs online so that they could complete it at home without any influence from competition. Before completing the test, prior information was provided: e.g., the name of the questionnaire, the scales being measured, and authors, among others. An email was provided in case of doubt when completing the questionnaire. Regarding the inclusion criteria, the referees belonged to the RFAF and had passed the corresponding physical tests to be able to referee their category during the season. In addition, to be able to participate in the study, the SRs must not have suffered serious injuries in the last 6 months that could affect them in the development of the season. It should be noted that no SR who participated in the study received any

compensation. We gave the opportunity, to those who wished, to know the results of the study after its publication.

## Statistical analysis

The SPSS 23.0 and AMOS 23.0 (IBM SPSS Statistic, Chicago, United States) programs were used to carry out the statistical analysis. First, to check the normality of the sample, the Kolmogorov–Smirnov test was carried out. Once the normal distribution of the sample was verified, descriptive tests and $t$-test were performed for age as independent sample. To compare different groups of experience and categories, one-way ANOVA test and subsequently a Bonferroni post hoc correction were performed to determine differences between the groups. The effect size in the ANOVA is presented by $\eta 2$ and was interpreted using the following criteria: small effect ($\eta 2 \leq .02$), moderate effect ($.02 < \eta 2 \leq .09$), and large effect ($\eta 2 > .09$) (*Lakens, 2013*). The effect size for the $t$ test was interpreted with Cohen's d values. For interpretation of the effect size, the following criteria were used: small effect ($d < .20$), moderate effect ($.20 \leq d < .80$), and large effect ($d \geq .80$) (*O'Donoghue, 2013*). Likewise, a Pearson r was performed to establish the relationships between the different dimensions of the REFS questionnaire with age, category, and seasons of experience of the SRs. The standards used for the classification of the correlation coefficients, established by *Hopkins (2000)*, were: trivial relationships ($r < .10$), small ($.10 < r < .30$), moderate ($.30 < r < .50$), large ($.50 < r < .70$), very large ($.70 < r < .90$), almost perfect ($r > .90$), or perfect ($r = 1$). Finally, a path analysis was performed to investigate the influence of referee characteristics and self-efficacy dimensions (game knowledge, decision making, pressure, and communication). To this end, the maximum likelihood method was selected with the bootstrapping procedure with 5000 iterations considering the violation of multivariate normality assumption (Mardia's coefficient $= 4.168$, $p < .01$) (*Kline, 2015*). The goodness of fit was judged with the following fit indexes: the rate $\chi 2$/degrees of freedom ($\chi 2/df$), the Comparative Fit Index (CFI), the Tucker–Lewis Index (TLI), the Standardised Root Mean Square Residual (SRMR), the Root Mean Square Error of Approximation (RMSEA) with its confidence interval at 90% (90% CI), the Akaike Information Criterion (AIC), and the Bayes Information Criterion (BIC). The rate $\chi 2/df$ is considered as an indicative of a good fit with values lower than 2, the comparative indexes (CFI and TLI) with values higher than 0.97, while the error of approximation indexes with values lower than 0.05 for SRMR and RMSEA (*Schermelleh-Engel, Moosbrugger & Müller, 2003*). AIC and BIC are typically used to compare the fit of competing models, where the model with the lowest AIC and BIC values would represent the best-fit model (*Kline, 2015*).

## RESULTS

Table 1 shows the results of the normality and reliability tests. The evaluated dimensions show medium-high reliability values. They present a normal distribution in all dimensions.

Table 2 shows the means, standard deviations, level of significance, and size of the effect of the following variables: knowledge of the game, decision making, pressure, communication, and global self-efficacy in baseline states. To assess the existence of differences between groups, Student's $t$-test was carried out for independent samples. Older SRs have higher

**Table 2  Comparison of Means of the Dimensions of Self-efficacy between SRs as a function of age.**

|  | Young SRs ($n = 156$) | Older SRs ($n = 100$) | $p$ | $d$ | |
|---|---|---|---|---|---|
| GK | $12.94 \pm 1.84$ | $13.64 \pm 1.29$ | .020 | .42 | M |
| DM | $12.56 \pm 1.78$ | $13.70 \pm 1.46$ | .000 | .68 | M |
| PR | $13.42 \pm 1.79$ | $14.26 \pm 1.34$ | .005 | .52 | M |
| CO | $16.77 \pm 2.08$ | $18.08 \pm 1.97$ | .001 | .64 | M |
| Σ Self-efficacy | $55.69 \pm 5.60$ | $59.68 \pm 5.17$ | .000 | .73 | M |

Notes.

GK, Game knowledge; DM, Decision making; PR, Pressure; CO, Communication; Σ Self-efficacy, Global self-efficacy; Young SRs, from 18 to 24 years-old; Older SRs, from 25 to 34 years-old.

Small effect (S) ($d < .20$), moderate effect (M) ($.20 \leq d < .80$), and large effect (L) ($d \geq .80$) (*O'Donoghue, 2013*).

**Table 3  Comparison of Means of the Dimensions of Self-efficacy between SRs according to the category.**

|  | National SRs ($n = 72$) | State SRs ($n = 50$) | Provincial SRs ($n = 78$) | Base SRs ($n = 56$) | $F(3,255)$ | $p$ | $\eta^2$ | |
|---|---|---|---|---|---|---|---|---|
| GK | $13.92 \pm 1.11$[4] | $13.16 \pm 1.43$ | $13.13 \pm 1.45$ | $12.46 \pm 2.37$[1] | 4.33 | .006 | .095 | L |
| DM | $13.86 \pm 1.27$[3,4] | $12.96 \pm 1.43$ | $12.77 \pm 1.91$[1] | $12.29 \pm 1.94$[1] | 5.15 | .002 | .111 | L |
| PR | $14.50 \pm .94$[4] | $13.76 \pm 1.88$ | $13.82 \pm 1.39$[4] | $12.68 \pm 2.06$[1,3] | 7.16 | .000 | .148 | L |
| CO | $17.97 \pm 1.91$ | $16.96 \pm 2.03$ | $17.05 \pm 2.04$ | $17.00 \pm 2.48$ | 1.81 | .149 | .042 | M |
| Σ; Self-efficacy | $60.25 \pm 3.71$[3,4] | $56.84 \pm 5.25$ | $56.77 \pm 5.28$[1] | $54.43 \pm 7.32$[1] | 6.34 | .000 | .133 | L |

Notes.

GK, Game knowledge; DM, Decision making; PR, Pressure; CO, Communication; Σ Self-efficacy, Global self-efficacy.

Effect size in the ANOVA: small effect (S) ($\eta^2 \leq .02$), moderate effect (M) ($.02 < \eta^2 \leq .09$), and large effect (L) ($\eta^2 > .09$) (*Lakens, 2013*).

levels in all dimensions of self-efficacy when compared as a global variable ($p < .05$) to the values of younger SRs, with moderate effect sizes ($d$).

The differences obtained between the categories are shown in Table 3. Regarding game knowledge, the SRs of the national category have the highest average; the values decrease across category until they reach the base SRs. However, there are only significant differences between the groups of national SR and base SR ($p < .05$). Regarding decision making, the national category had the highest values, and the values progressively decrease in lower categories; there are significant differences between the national SRs with the provincial SR and base SR ($p < .05$). In the dimension of preassure, national SRs displayed a higher level of control than the base SRs. There were significant differences between the national and base SRs ($p < .05$) and between the provincial SRs and base SRs ($p < .05$). Regarding communication, the national SRs had a higher mean value, but there were not significant differences between different groups. Finally, SRs of higher category had higher mean global self-efficacy scores, and the scores decreased progressively with significant differences in national SRs; there was also a significant difference between Provincial and base SRs ($p < .05$).

Table 4 shows the results of self-efficacy based on experience using the one-way ANOVA test. For game knowledge, we observed a higher score in SRs with more experience in refereeing. Significant differences were seen between the group of SRs with less experience and the experienced and highly experienced SRs ($p < .05$), but there was no difference between the experienced and highly experienced SRs. The mean decision
**Table 4  Mean comparisons of the self-efficacy dimensions between SRs based on experience.**

| | Less-Experience SRs (n = 80) | Experience SRs (n = 106) | High-Experience SRs (n = 70) | F(2,255) | p | η² | |
|---|---|---|---|---|---|---|---|
| GK | 12.40 ± 2.10[2,3] | 13.45 ± 1.40[1] | 13.77 ± 1.14[1] | 7.97 | .001 | .113 | L |
| DM | 12.05 ± 1.87[2,3] | 13.28 ± 1.73[1] | 13.69 ± 1.11[1] | 10.69 | .000 | .146 | L |
| PR | 13.00 ± 1.95[2,3] | 13.87 ± 1.59[1] | 14.43 ± 1.04[1] | 7.77 | .001 | .111 | L |
| CO | 16.48 ± 2.28[3] | 17.30 ± 2.06 | 18.17 ± 1.69[1] | 6.46 | .002 | .094 | L |
| Σ; Self-efficacy | 53.93 ± 6.49[2,3] | 57.91 ± 5.13[1] | 60.06 ± 3.68[1] | 13.35 | .000 | .176 | L |

Notes.

GK, Game knowledge; DM, Decision making; PR, Pressure; CO, Communication; Σ Self-efficacy, Global self-efficacy; Less-Experience SRs, 1–4 years/seasons; Experience SRs, 5–8 years/seasons; High-Experience SRs, >8 years/seasons.

Effect size in the ANOVA: small effect (S) ($\eta^2 \leq .02$), moderate effect (M) ($.02 < \eta^2 \leq .09$), and large effect (L) ($\eta^2 > .09$) (Lakens, 2013).

**Table 5  Correlation between Age, Category and years of experience with knowledge of the game, decision making, ability to withstand pressure, communication and one-dimensional Self-efficacy.**

| | Age | Category | Experience |
|---|---|---|---|
| GK | .255[**] | .295[**] | .314[**] |
| DM | .315[**] | .323[**] | .382[**] |
| PR | .303[**] | .353[**] | .364[**] |
| CO | .313[**] | .164[*] | .291[**] |
| Σ Self-efficacy | .374[**] | .347[**] | .421[**] |

Notes.

GK, Game knowledge; DM, Decision making; PR, Pressure; CO, Communication; Σ Self-efficacy, Global self-efficacy.

[*] $p < 0.01$.

[**] $p < 0.001$.

making progressive increased with increasing experience in refereeing, and there were significant differences between SRs with less experience and with the experienced and highly experienced SRs ($p < .05$). However, between the experienced and highly experienced RH groups, no significant differences were observed. RP follows the same pattern as the previous variables with a progressive increase as the SR acquires more experience. There were significant differences between the less experienced SR group and the experienced and very experienced SRs ($p < .05$), but there was no difference between the experienced and highly experienced SRs. There was an increase in the mean communication score with increasing experience, with significant differences between the less experienced and highly experienced RH groups ($p < .05$). The global self-efficacy score increased as refereeing experience increased. Likewise, significant differences were found between SRs with less experience and the experienced and highly experienced SR ($p < .05$).

Table 5 shows the correlations of the REFS questionnaire variables with age, category, and years of experience. All dimensions of self-efficacy have positive correlations with age, category, and experience. The highest correlations were found in global self-efficacy, with coefficients ranging from .35 and .42. decision making had the second highest correlation with age and experience ($r = .32$ and $.38$; $p < .01$; respectively).

Finally, multiple path analysis was performed in order to test the effect of intervening variables. In Fig. 1, the path analysis of the influence of all the age, category and experience with dimensions of self-efficacy can be observed. Table 6 presents the fit index obtained by

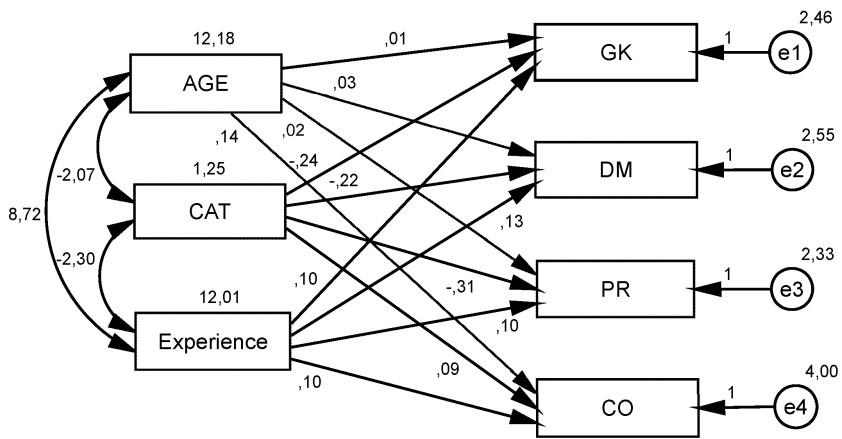

**Figure 1** **Path Analysis with unstandardized coefficients between age, experience and category with self-efficacy dimensions.** GK, Game knowledge; DM, Decision making; PR, Pressure; CO, Communication; CAT, Category.

**Table 6** **Goodness-of-fit measures obtained in the factor model tested for the age, category and experience with multidimensional self-efficacy.**

| $\chi^2$/DF | CFI | GFI | RMR | RMSEA | RMSEA (90% IC) | AIC | BIC |
|---|---|---|---|---|---|---|---|
| 2.914 | .936 | .941 | .040 | .048 | .036–.054 | 1958.40 | 1961.36 |

the factor model. This model obtained a good fit to the observed data and the examination of standardized regression weights revealed some characteristics in its internal structure for the model (see Fig. 1).

## DISCUSSION

The aims of this study were to demonstrate that self-efficacy is greater in SR of higher category, age and experience (measured as seasons as a referee), and to determine the relationship between these variables. All the analyses carried out have served to find variability in the self-efficacy of a group as homogeneous as the referees. Specifically, in this study it is intended to know the profile of self-efficacy that SRs have, so it is necessary to know in depth how the factors that correspond to their characteristics affect. The results showed that SRs of higher category have a higher level of self-efficacy, coinciding with a study carried out in handball referees (*Diotaiuti et al., 2017*). In the game knowledge, the SRs of the National category obtained significantly higher average scores compared to the other categories. This may be due to the domain of the regulations and to the demanding theoretical tests Referee Committees require, which increase in complexity across categories, adding to SRs' awareness of rules of the game (*Ittenbach & Eller, 1988*). The National category SRs presented significantly higher scores in decision making than the Provincial and Base SRs, possibly due to the strong relationship with game knowledge (*Guillén et al., 2019*; *Myers et al., 2012*). Increased game knowledge enables greater certainty in decision making in competition, in the application of regulations, and in the swift
translation of judgement into actions (*Helsen & Bultynck, 2004*; *MacMahon et al., 2007*). In communication no differences were found between the categories, suggesting lack of a relationship (Table 5). This may be because SRs are aware that effective communication is essential to maintain control of the game (*Grunska, 1999*). Confidence and certainty in communicating both verbally and nonverbally is emphasized starting in the base category. This may explain why there are no differences between categories, in addition to the fact that in lower categories, the players are small and the SRs progressively acquire and improve their language as the Committee gives them confidence by assigning them higher category matches. Preassure is higher in national SRs, since they have a greater impact in higher categories and should evolve along with psychological skills. These findings support the hypothesis (H1a) that higher-category SRs have higher levels of self-efficacy. Regarding age, older SRs had higher levels of self-efficacy, coinciding with the recent study by *Karacam & Adiguzel (2019)* that used the same questionnaire (REFS) in basketball referees; this study found moderate and even high positive relationships in variables such as age. Our results differ from those of *Nazarudin et al. (2014)* in rugby referees; this study found no significant differences between the same age groups. This may be explained by the older age of the sampled referees, with only 25% of the sample being less than 30 years and with a mean age of 33.4 $\pm$1.5 years. In addition, a study in SRs from the north of Spain found that those over 40 years of age have greater stress control, which is positively related to self-efficacy (*González-Oya & Dosil, 2004*). For these reasons, hypothesis H1b is supported.

High-experienced SRs had higher levels of self-efficacy, corroborated with large effect sizes and agreement with results obtained by *Diotaiuti et al. (2017)*. In addition, the correlation results are similar those of *Karacam & Adiguzel (2019)*. However, they are much higher than those found in the study by *Karaçam & Pulur (2017)* in which small relationships were established ($0.1 < r < 0.3$) in the same dimensions of the REFS questionnaire with experience, in addition to age, which could be due to the fact that the authors incorporated SR with basketball and handball referees in the same analysis, hypothesizing that the requirements at psychological levels and more specifically self-efficacy for each sport and, in categories of the sport itself, may not be the same. Regarding decision making, SRs with less experience are perhaps more cautious when making decisions since they have less control of the regulation and do not want to risk making a wrong decision. This is supported by *Myers et al. (2012)*, who found a positive relationship between decision making and game knowledge (0.80). This may also be due to the lack of communication skills in less experienced SRs who may show greater insecurity when addressing players and coaches, added to a nonverbal language conveying SR insecurity to fans. RP management is higher in SRs with more experience since their psychological skills are more developed and they have lower levels of stress than those with less experience (*González-Oya & Dosil, 2004*; *Soriano Gillué et al., 2018*). Communication dimension of SR self-efficacy evolves over time and experience; at the beginning, it may be normal for SRs to not know how to address technicians or players safely or to insecurity through nonverbal language to fans. However, as observed, this aspect, like the others, evolves over years of experience. Finally, as in our study, the increase in self-efficacy in terms of experience was also reflected in adolescents who participated in an extracurricular sports

activity program in the study by *Reverdito et al. (2017)* where, through a linear regression, students with more than two years of experience obtained greater benefits in terms of self-efficacy compared to those with less experience. Due to the absence of findings on the relationship of self-efficacy and experience in the field of refereeing, we justify the results obtained in this study with those found in the adolescent population of the study by *Reverdito et al. (2017)* as they are in line with our study. Therefore, we can confirm the H1c and H2 hypotheses that high-experienced SRs have higher self-efficacy and that there are moderate positive correlations between self-efficacy and the age, category, and experience of the SRs, respectively.

Low levels of self-efficacy are related to greater stress and anxiety (*Guillén et al., 2019*), which could cause burnout in SRs and trigger the abandonment of sports practice. In addition, optimal self-efficacy could result in less impulsive decision making due to the absence or control of stress over time and alleviation of negative influence on the perception of the environment, body, time, cognition, and memory (*Tornero-Aguilera, Robles-Pérez & Clemente-Suárez, 2017*). This would enable the SRs to collect more information about each play and make better decisions. We consider self-efficacy to be of great importance in SR performance, and according to *Garcés de los Fayos & Vives (2003)*, this has been scarcely studied. There are few psychological intervention programs in SR; in Spain, there is one study (*Ramírez et al., 2006*) with the widely accepted implementation of the PHIPA Program ("Intervention Program in Psychological Skills in Refereeing"). Currently, in Spanish 3rd and 2nd B category referees the Talent and Mentors program mentioned in the SR study by *Fernández-Elías, Gómez-López & Clemente-Suárez (2017)* focuses mainly on the aspects of sports performance and regulations and their application. Considering the results of this study and others, SRs of lower categories, ages, and experience who have lower levels of self-efficacy and probably other psychological abilities could benefit from programs such as PHIPA or the of Talents and Mentors to perfect refereeing from the base and develop skills as new SRs, thus preventing burnout produced by verbal or physical aggressions that lead to abandonment of sports practice (*Alonso-Arbiol et al., 2005*).

## Limitations

The present study shows different limitations. First, it is a cross section and the type of sampling it presents is non-probabilistic. Second, the use of self-reported measures that may be subject to a social desirability bias and that could influence the results. Another limiting factor can also be the small number of the sample, although, it is representative of the amateur SR. Finally, it would be necessary to include the physical condition values in order to establish the term level within the factors that predict self-efficacy. Fitness tests are used to move up or down in competition by causing a referee category. In this article, the category has been taken into account, but there is a physical condition that could differ within the same category. For these reasons, the term level in the referee could be very interesting for the future.

Future research should study the relationship between self-efficacy and impulsivity of SRs since greater self-efficacy and less stress is correlated with improved perception of the environment and judgment, which may influence confidence in impulsive decisions. These

do not imply that they are incorrect, but that they may take greater risks. This study could also be extrapolated to other team sports (basketball, volleyball, and handball) in which the questionnaire can be used to determine whether results are similar or for each sport and whether certain psychological skills are required. Another possibility for future study could lie in conducting longitudinal cross-sectional research with a Bayesian approach, assessing the evolution of SR self-efficacy during the season at two or three different times, or even viewing over 2 or more seasons as the referee's self-efficacy evolves. You could also compare and study the differences between professional and amateur SRs and see if there are differences in terms of self-efficacy. As a final contribution to future studies, control of stress or stress in SR could be evaluated together with self-efficacy in order to establish relationships between stress and self-efficacy in SR.

## CONCLUSIONS

The main findings of the study show that self-efficacy of the amateur SR is conditioned by category or level of the competition, age, and experience. These data open a line of research on self-efficacy around SR due to the influence (determined by a large effect size) of SR characteristics such as age, category and experience. The data obtained show the profile of the amateur SR in terms of self-efficacy between different categories. In addition, this self-efficacy could affect the decision-making and other decisive behaviors in the competition. In terms of practical applications, they could help understand and prevent burnout in lower-category SRs, age and experience, at the same time that it could be a cause of why certain higher-category SRs do not perform as well as in lower categories. Therefore, this study provides relevant information for refereeing institutions, recommending promoting psychological intervention programs to develop better psychological skills of newer, lower-category, and younger SRs to increase their self-efficacy and their relationship with decision making as well as to prevent burnout and abandonment of refereeing practice.

### Funding
This study has been funded by the project PPJIA2020.04, of the Precompetitive Research Projects program for Young Researchers of the Own Plan 2020, of the University of Granada. The funders had no role in study design, data collection and analysis, decision to publish, or preparation of the manuscript.

### Grant Disclosures
The following grant information was disclosed by the authors:
Precompetitive Research Projects program for Young Researchers of the Own Plan 2020, of the University of Granada: PPJIA2020.04.

### Competing Interests
The authors declare there are no competing interests.

## Author Contributions

- José López Aguilar performed the experiments, analyzed the data, authored or reviewed drafts of the paper, and approved the final draft.
- Alfonso Castillo-Rodriguez conceived and designed the experiments, performed the experiments, analyzed the data, authored or reviewed drafts of the paper, and approved the final draft.
- José L. Chinchilla-Minguet performed the experiments, prepared figures and/or tables, and approved the final draft.
- Wanesa Onetti-Onetti conceived and designed the experiments, performed the experiments, prepared figures and/or tables, and approved the final draft.

## Human Ethics

The following information was supplied relating to ethical approvals (i.e., approving body and any reference numbers):

This study was approved by the Ethics Committee of the University of Granada, Spain (Ethical Application Ref: 471/CEIH/2018).

## Data Availability

Raw data are available as a Supplementary Files.

## Supplemental Information

Supplemental information for this article can be found online at http://dx.doi.org/10.7717/peerj.11472#supplemental-information.

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
