# Peer review of "Relationship between age, category and experience with the soccer referee’s self-efficacy"

_PeerJ, doi:10.7717/peerj.11472_

## Round 0.1 · original submission · Major Revisions

Dear Drs. López-Aguilar and colleagues:

Thanks for submitting your manuscript to PeerJ. I have now received three independent reviews of your work, and as you will see, the reviewers raised some concerns about the research. Despite this, these reviewers are optimistic about your work and the potential impact it will have on research studying sports performance and prevention of sports abandonment with emphasis on referees. Thus, I encourage you to revise your manuscript, accordingly, taking into account all of the concerns raised by both reviewers.

The reviewers provide many concerns about the introduction, contribution of the study, data analysis, type of sampling method, limitation and implications; all of these issues must be tackled and justified in your revision. Please especially provide a clearer justification for your research design and findings.

Please use the comments by the reviewers to add missing information where possible. Try to restructure your manuscript for clarity, avoiding redundancy and streamlining sections for effective delivery.

Therefore, I am recommending that you revise your manuscript, accordingly, taking into account all of the issues raised by the reviewers. I do believe that your manuscript will be greatly improved once these issues are addressed.

Good luck with your revision,

-joe

Reviewer 1 ·

Basic reporting

The manuscript is well-written in English.

The structure for the introduction section needs to be improved to gain clarity and rationale, although background and context provided is sufficent. However, the contribution the study is missed and litarature references needs be revised.

The article follows the basic structure for any research work: Introduction, Materials and Methos, Results, Discussion and Conclusions. Nevertheless, a limitation section must be included and implications for practice should be improved. The different tables are showed in an adeaquate and clear way.

The results obtained responded to the objectives and hipotheses raised.

Experimental design

The study represents an original research.

Although the research questions were well defined, the contribution of the research is missed and needs to be urgently addressed.

The reserach meets all ethical standards by having their approval of the Ethical Committe of a particular university.

Method was suitably described, providing detailed information to its future replication.

Validity of the findings

Novelty of results is difficult to know by not specifying the contribution of the study to the scientific knowledge. However, the noexpected results were well discussed and justified. The benefits derived from this research should be reforced.

Overall, the data analysis were adequate, although a serie of points related to missing values, outliers and path analysis instead of linear regression analysis must be tackled and justified by the authors.

The conclucions of the study responded to the objectives raised for the study. Nonetheless, they should be presented in a more detailed way.

Additional comments

Introduction
General comment.
In my viewpoint, the structure of the introduction section needs to be largely improved to gain clarity, consistency and rationale. It is really difficult to follow the reading with respect to their ideas. The authors presented stress and its relationships with other psychological outcomes, making me think that this variable would then be analysed, but it did not do so. Similarly, they accurately described the Referee Self-Efficacy Scale as a valid and reliable measure of self-efficacy, but the study did not aim to examine its psychometric properties. The conceptualisation of general self-efficacy and referee self-efficacy also needs be to enhanced. In addition, the contribution and gap of the study are missed.

Specific comments:
Comment 1: Line 49 and 50: To justify the little body of research focused on referees, there is a need to use a more current study than the Guillen’s (2003) one. Since 2003, the body of research could be increased.

Comment 2: Line 56: Please, swap “i.e.” for “such as”.

Comment 3: There is no point in talking about stress. It is advisable to draft this part again. Indeed, stress could be a consequence associated with a low level of self-efficacy in referees.

Comment 4: Lines 79 – 86. It is confusing for me. The ideas do not sound good. First, the concept of general self-efficacy must be improved because some key words are missed: a set of beliefs or specific context or task. Given that the self-efficacy is inherent to a specific activity, Bandura advocates for distinguishing different types of self-efficacy, including referee’s self-efficacy. After conceptualising general self-efficacy, it is more logical to find the definition of referee self-efficacy (higher-order factors) and then its four dimensions: game knowledge, decision making, pressure and communication (primary-order factors). Next, it is recommendable to show the different associations of referee self-efficacy with different adaptive or maladaptive outcomes derived from arbitrating a match. Thus, there is no point in talking about self-efficacy in athletes due to they were not studied in this research (lines 80-82). Indeed, as the authors know there is a big controversy about the consideration of referees as professional sportspeople.

Comment 5: In my opinion, there is no point in showing the full version of the Referee Self-Efficacy Scale because the objective of this research was not to analyse the instrument’s psychometric properties. Thus, the information about the instrument is better to find it in the instrument section.

Comment 6: Introduction: Lines 96 – 102: It is unnecessary to present and describe the six key variables for refereeing success given that they were not examined. This detailed description makes me think that the authors also wanted to study them.

Comment 7. Introduction: The contribution and gaps of the study are missed. Which is the contribution of the study to the current scientific knowledge? Is it important to study the football soccer’ sense of efficacy? Would there be differences between football referees and other sports ones?

Comment 8. Introduction: Line 120: González-Oya & Dosil (2004) -> González-Oya and Dosil (2004)

Materials and Method
Overall, the method followed and materials were well described.

Specific comments:
Comment 1: Participants: I suggest including the maximum and minimum values for age, weight, height, and experience.

Comment 2: Participants: Line 137: The concept of data baseline contrasts to the cross-sectional design used in the study (line 156). In my viewpoint, data baseline suggests the existence of a post-test measure.

Comment 3: Participants: The sampling method is missed. It is therefore recommendable to specific how the participants were recruited and selected.

Comment 4. Instruments: Specify the numbers of items for each factor comprising the instrument. It is also recommended to include a sample item of each of the four factors.

Comment 5. Instruments. Lines 150-153. In my opinion, the results given on the Cronbach’s alpha coefficients would be better ubicated in the results section. Additionally, it should be considered that the research specialised in psychometry establishes that Cronbach’s alpha values equal to .80 or higher must be obtained when comparing mean differences among groups (Nunnally & Bernstein, 1994; Viladrich et al., 2017).

Comment 6. Procedure: I suggest swapping “Procedure” for “Design and Procedure” by tackling the type of design adopted in this section too.

Comment 7. Procedure: Although the questionnaire was administrated via online, I wonder if the survey respondents had any type of information for their completion. It would also be recommended to report if the participations received any type compensation for filling it and the average time spent for their administration.

Comment 8. Data analysis. Prior to normality assumption, it would be mandatory to inform about missing values and univariate and multivariate outliers.

Comment 9. Data analysis: There is no information about a linear regression analysis in the data analysis section, although their findings were displayed in the results section and Table 6. Even I wonder if a structural equation modelling analysis with latent (self-efficacy) and observed (age, experience, etc) variables had been better option than the linear regression analysis (Kline, 2015).

Results

Comment 1. Table 1. Please, specify p-value associated with the Z statistic used in the Kolmogorov–Spearmon test.

Comment 2. Table 2. Which does M mean found on the right side of the table?

Comment 3. Table 3. Which do G and M mean found on the right side of the table?

Comment 4. Table 4. Which does G mean found on the right side of the table?

Comment 5. Line 197 and following times: “one-dimensional variable”. In psychometric terms, an one-dimensional variable is the one measured by a single factor (Kline, 2015). However, referee self-efficacy is here measured by the estimation of four primary-order factors. This makes that referee self-efficacy a higher-order factor, indicating a global referee self-efficacy score. Please, be careful about the expressions.

Discussion
The discussion is well structured. In general terms, the main results are discussed, contrasted with previous studies and the justifications proposed for these findings are logical and consistent.

Comment 1. Line 275 and 286: Karakam & Adiguzel (2019) -> Karakam and Adiguzel (2019)

Comment 2. Line 312: Fayos & Vives (2003), -> Fayos and Vives (2003),

Comment 3. It is a mandatory requirement to consider a limitations section, including the main limitations derived from the study. For instance: the cross-sectional design, the type of non-probabilistic sampling method or the use of self-reported measures are some limitations that this study had.

Comment 4. It is important to underline the practical implications of this study. The conclusion section showed poor implications for practice. Indeed, the conclusion section could be largely improved.

The study references
Please, ensure that references have not got types. Below, the authors can find some typos detected.
Comment 1: Lines 398- 400: Check this reference because there are typos.

Comment 2: Lines 405: science: a -> science: A

Comment 3: Line 423: Effi-cacy. Please, check this out.

List of references
Kline, R. B. (2015). Principles and practice of structural equation modeling (4th ed.). The Guilford Press.
Nunnally, I. H., & Bernstein, J. C. (1994). Psychometric theory (3rd ed.). McGraw‐Hill.
Viladrich, C., Angulo-Brunet, A., & Doval, E. (2017). A journey around alpha and omega to estimate internal consistency reliability. Annals of Psychology, 33(3), 755–782. https://doi.org/10.6018/analesps.33.3.268401

Reviewer 2 ·

Basic reporting

1. The use of the term “category” to reference a referee’s level based on the standard of soccer they can referee is unclear until line 106 after three references (including the abstract). It is recommended that the “category” is altered to “level” or further clarification is made in order to improve understanding.
2. Your review of self-efficacy studies on referees is concise but more justification could be added to the knowledge gap being filled by your study between lines 103-130.
3. It is suggested that brief clarification is added to improve layperson understanding to specifical clarify why “optimum levels of self-efficacy are important” in the abstract on line 28-30.
4. Further clarification on the “psychological skills” mentioned on lines 60 and 63 would improve clarity.
5. The sentence on lines 32-33 “Two-hundred fifty….” Could be improved for clarity. Adding “completed” before “Referee Self-efficacy Scale….” Would suffice
6. Although abbreviations are used well throughout, “RH” on line 313 and “AB Referees” need explaining to improve comprehension.
7. Although the connection is appropriate, a better link between a referee and arbitrator could be made on line 84 in order to improve clarity.

Experimental design

1. Further details and context are required about the SR participants outline on line 133. The first indication that these referees were voluntary/amateur is made in the conclusion. As stated, this could have an impact on the findings so should be outlined earlier alongside further contextual information. For instance, what is a national category voluntary referee from Andalusia equivalent to in the professional game?
2. Furthermore, are all SR adjudicating the same age group of participants? You may suggest this on line 270 stating, “the players are small”. A difference in the age group a SR is adjudicating could have a big impact on their “certainty in communicating…”.
3. Providing further detail about the validity of the Referee Self-Efficacy Scale. Meyers et al (2012) provided robust construct validity for the measure and Cronbach’s Alpha does not solely represent this. For further guidance, see Flake and Fried’s (2020) ‘Measurement Schmeasurement: Questionable Measurement Practices and How to Avoid Them’.
4. I suggest refining the research questions and aims of the study to improve comprehension and clarity. The study aims outlined in the introduction bare no reference to the “endogenous characteristics” stated on lines 254-255 of the discussion.
5. Further justification is required to outline the reason why the study was conducted in March 2020 and the implications that may have on the study.
6. A suggestion for future research following your study would be to adopt more longitudinal research methods and applying a Bayesian approach to data analysis.

Validity of the findings

1. I commend the authors for their robust data analysis and attempts to validate their findings through multiple statistical tests. Thank you for providing the raw data set and for outlining your results clearly in the manuscript. If there is a weakness it is tying the findings and concluding remarks back to the research question and the gap in knowledge that has been suggested above.
2. You should add more support for the reason behind conducting the number of statistical tests you have. It could be misconstrued that you have gotten lost down the “garden of forking paths” and it would help to show your rationale.

Additional comments

I commend the author for a very interesting paper and their attempt to bridge a much-needed gap in the literature. With clearer justification for your research design and findings, you have an excellent paper that will contribute to the discussion on self-efficacy literature in sport.

Reviewer 3 ·

Basic reporting

no comment

Experimental design

no comment

Validity of the findings

no comment

Additional comments

Comments to the Authors:
The purpose of this article was to examine referee self-efficacy in Spanish referees at a variety of age, competence, and experience levels. Overall, I found the paper interesting, although somewhat hard to follow at times. I have provided some comments below based on my multiple reads of the paper.

Main Points:
The authors conducted many statistical tests. While I don’t see that as an explicit problem, as a reader if was difficult for me to discern what tests are most appropriate. I would strongly suggest for the authors to think critically about their main research questions, frame the introduction around those research questions, and discuss the findings tied to those questions. As a reader, I didn’t always know how a statistical test was tied to something that the authors set up in the introduction. I think it is worth removing some of the tests that add little to the main research question (perhaps the regression at the end).

This point follows the previous point. I would strongly encourage the authors to list out all of the hypotheses they plan on testing in the introduction. As a reader, I was surprised that there was a regression included in the analysis because it was not mentioned in the introduction at all. If you are going to include the statistical test, it should be clearly derived from the introduction that was written.

I would also like the authors to strongly consider sticking with consistent terms throughout the paper. The authors used the term referee and arbitrator throughout the manuscript. It can be difficult for readers to make sure you know what you are discussing. Please stick with a term and use it throughout (even if it seems repetitive).

Remove Acronyms. I am fine with soccer referee (SR) being shortened, but many times during the paper I had to go back to see what the subscale acronyms stood for. For readability, make sure to write out the full term.

Minor issues:
L 1- The authors never make clear the distinction about the controllable variables in the manuscript. The title should be revised to match what the authors did.

L 28-29- The first sentence reads awkward to me. Perhaps change to “Soccer referees encounter stressful situations during…”

L 34- There should be a hyphen between fifty-six

L 35- Change “carried-out” to administered

L 40- Remove the term influence. Everything that was done in the study was cross sectional and correlational. Perhaps use relate

L 50-51- In my experience, referees work at sports games, not sports practice

L 58- Soccer referees do not control the behavior of athletes.

L 62- When you state travel distances, I believe you mean run.

L 66- Remove the phrase “face of the permanent”

L 73- Is this the finding for just one position (soccer forwards)? It seems like it wouldn’t be just for soccer forwards.

L 74-75- This sentence is unclear to me. Please rephrase.

L 103- The variables listed there are more accurately described as subscales

L 113-115- This sentence is unclear to me. Are the referees athletes? I think a rephrase here is necessary

L 158- What do you mean by an ad hoc instrument?

L 274 and onward- Once you restructure the introduction and make clear ties to the hypotheses you test, I believe you need to restructure some of the discussion to make clear the new knowledge generated.

L 503- The term “Grown-up SRs” does not appropriately describe the referees in this sample. Use a different term.

Table 4. The term “non-experience SRs” does not describe the referees in the sample well. Find an alternative term.

Table 5. How did you create a value for category? I might have missed it in the manuscript, but it was unclear in this table.

---

## Round 0.2 · Minor Revisions

Dear Drs. López-Aguilar and colleagues:

Thanks for revising your manuscript. The reviewers are mostly satisfied with your revision (as am I). Great! However, there are a few remaining issues that need attention. Please address these ASAP so we may move forward with your manuscript.

Please describe the path analysis in more detail, providing necessary information (particularly for repeatability) and a goodness-of-fit measure for assessment. Figure 1 has some issues per Reviewer 1 that need your attention.

Therefore, I am recommending that you revise your manuscript, accordingly, taking into account these issues. I do believe that your manuscript will be greatly improved once these issues are addressed.

Good luck with your revision,

-joe

Reviewer 1 ·

Basic reporting

The authors have completed a good job by improving the introduction, method, discussion and conclusions. In my view, the quality of the manuscript has been meaningully increased. However, I am seriously concerned about path analysis made and showed by the authors.

Experimental design

The authors have meaningfully improved this section of the manuscript. Congrats.

Validity of the findings

Please, provide information about path analysis and Figure 1. Path analysis is a type of structural equation modeling (Kline, 2015) performed using AMOS, Lisrel, EQS, Mplus or R. This type of analysis is more poweful one than linear regression analysis. In fact, path analysis had to replace linear regression analyses made. In this same vein, the hypothetised factor model in terms of path analysis provides goodnees-of-fit measures that should be interpreted according to cuf-off points established by previous psychometry studies (e.g., Hu and Bentler, 1999 or Marsh et al., 2004). It is also needed to gather evidence on multivariate normality assumption (e.g., Mardia's coefficient) and to detail estimation method.
Regarding Figure 1. Typically, manifest variables are represented by rectangles instead of circles. Circles are used to represent latent variables (or latent factors). In addition, R2, total variance explained, should be for dependent (or consequence) variables instead of independent (or predictive) variables, to the extent that R2 reflects the porcentage that indenpent variables explain for each dependent variable. Latly, the representation of lantent variable G-SE generates serious doubts to me. Figure shows that G-SE (high-order factor) predicts four primary-order factors, when the high-order factor had to be predicted by the four primary-order factors. Indeed, it is totally unnecesary to include G-SE as high-order factor by being already considered as manifest variable. Anyway, more justification and rationale is needed to gain clarification on this point.

References

Arbuckle, J. L. (2011). IBM® SPSS® AMOSTM version 20.0. IBM SPSS.
Hu, L., & Bentler, P. M. (1999). Cutoff criteria for fit indexes in covariance structure analysis: Conventional criteria versus new alternatives. Structural Equation Modeling, 6(1), 1–55. https://doi.org/10.1080/10705519909540118
Kline, R. B. (2015). Principles and practice of structural equation modeling (4th ed.). The Guilford Press.
Marsh, H. W., Hau, K.-T., & Wen, Z. (2004). In search of golden rules: Comment on hypothesis-testing approaches to setting cutoff values for fit indexes and dangers in overgeneralizing Hu and Bentler’s (1999) findings. Structural Equation Modelling, 11(3), 320–341. https://doi.org/10.1207/s15328007sem1103

Additional comments

I would like to thank the authors for their great job in providing a detailled response to every comment and suggestion proposed. However, I am seriously concerned about path analysis given that it does not followed the habitual procedure.

Reviewer 3 ·

Basic reporting

no comment

Experimental design

no comment

Validity of the findings

no comment

Additional comments

I believe the changes to the manuscript made it a significantly better final product. Thank you for taking the time to consider each comment.

---

## Round 0.3 · Minor Revisions

Dear Drs. López-Aguilar and colleagues:

Thanks for again revising your manuscript. There are some issues to address with your statistical analyses. Please address these per the concerns raised by reviewer 1. I do believe that your manuscript will be ready for publication once these issues are addressed.

Good luck with your revision,

-joe

Reviewer 1 ·

Basic reporting

No additional information is required.

Experimental design

No additional information is required.

Validity of the findings

Additional information about path analysis is required.

Additional comments

The authors continue to provide no information about how path analysis was run. Given that they used AMOS, they should inform on the multivariant normality assumption through Mardia's coefficient (Byrne, 2016). The estimation method is also missed. Additionally, the 5000-resampling bootstrapping technique would be applicable in the case of violation of the multivariant normality assumption (Byrne, 2016). In addition, assessment of goodness of fit should be reported by a variety of fit indexes. For AMOS: chi squared/ degree of freedom, CFI, IFI, TLI, SRMR and RMSEA (90%CI). Following the previous research, the chi squared/ df is adequate with values less than 5 and excelecent with values below 3. CFI, TLI and IFI are adequate with values higher than .90 and excelent with values over .95. SRMR and RMSEA are adecuate with values below .080, while values lower than .06 are excelent for RMESEA (Hu & Betler, 1999; Marsh et al., 2004). All of this information is missed.

---

## Round 0.4 · accepted · Accept

Dear Drs. López-Aguilar and colleagues:

Thanks for revising your manuscript based on the concerns raised by the reviewer. I now believe that your manuscript is suitable for publication. Congratulations! I look forward to seeing this work in print, and I anticipate it being an important resource for groups studying sports performance and prevention of sports abandonment with emphasis on referees. Thanks again for choosing PeerJ to publish such important work.

Best,

-joe